# Multi-Layer Feature Reduction for Tree Structured Group Lasso via Hierarchical Projection

**Jie Wang**[1], **Jieping Ye**[1,2]
[1]Computational Medicine and Bioinformatics
[2]Department of Electrical Engineering and Computer Science
University of Michigan, Ann Arbor, MI 48109
{jwangumi, jpye}@umich.edu

## Abstract

Tree structured group Lasso (TGL) is a powerful technique in uncovering the tree structured sparsity over the features, where each node encodes a group of features. It has been applied successfully in many real-world applications. However, with extremely large feature dimensions, solving TGL remains a significant challenge due to its highly complicated regularizer. In this paper, we propose a novel **M**ulti-**L**ayer **F**eature **re**duction method (MLFre) to quickly identify the inactive nodes (the groups of features with zero coefficients in the solution) hierarchically in a top-down fashion, which are guaranteed to be irrelevant to the response. Thus, we can remove the detected nodes from the optimization without sacrificing accuracy. The major challenge in developing such testing rules is due to the overlaps between the parents and their children nodes. By a novel hierarchical projection algorithm, MLFre is able to test the nodes independently from any of their ancestor nodes. Moreover, we can integrate MLFre—that has a low computational cost—with any existing solvers. Experiments on both synthetic and real data sets demonstrate that the speedup gained by MLFre can be orders of magnitude.

## 1 Introduction

Tree structured group Lasso (TGL) [13, 30] is a powerful regression technique in uncovering the hierarchical sparse patterns among the features. The key of TGL, i.e., the tree guided regularization, is based on a pre-defined tree structure and the group Lasso penalty [29], where each node represents a group of features. In recent years, TGL has achieved great success in many real-world applications such as brain image analysis [10, 18], gene data analysis [14], natural language processing [27, 28], and face recognition [12]. Many algorithms have been proposed to improve the efficiency of TGL [1, 6, 11, 7, 16]. However, the application of TGL to large-scale problems remains a challenge due to its highly complicated regularizer.

As an emerging and promising technique in scaling large-scale problems, *screening* has received much attention in the past few years. Screening aims to identify the zero coefficients in the sparse solutions by simple testing rules such that the corresponding features can be removed from the optimization. Thus, the size of the data matrix can be significantly reduced, leading to substantial savings in computational cost and memory usage. Typical examples include TLFre [25], FLAMS [22], EDPP [24], Sasvi [17], DOME [26], SAFE [8], and strong rules [21]. We note that strong rules are *inexact* in the sense that features with nonzero coefficients may be mistakenly discarded, while the others are exact. Another important direction of screening is to detect the non-support vectors for support vector machine (SVM) and least absolute deviation (LAD) [23, 19]. Empirical studies have shown that the speedup gained by screening methods can be several orders of magnitude. Moreover, the exact screening methods improve the efficiency without sacrificing optimality.

However, to the best of our knowledge, existing screening methods are only applicable to sparse models with simple structures such as Lasso, group Lasso, and sparse group Lasso. In this paper, we

propose a novel **M**ulti-**L**ayer **F**eature **re**duction method, called MLFre, for TGL. MLFre is *exact* and it tests the nodes hierarchically from the top level to the bottom level to quickly identify the inactive nodes (the groups of features with zero coefficients in the solution vector), which are guaranteed to be absent from the sparse representation. To the best of our knowledge, MLFre is the first screening method that is applicable to TGL with the highly complicated tree guided regularization.

The major technical challenges in developing MLFre for TGL lie in two folds. The first is that most existing exact screening methods are based on evaluating the norm of the subgradients of the sparsity-inducing regularizers with respect to the variables or groups of variables of interests. However, for TGL, we only have access to a *mixture* of the subgradients due to the overlaps between parents and their children nodes. Therefore, our first major technical contribution is a novel hierarchical projection algorithm that is able to exactly and efficiently recover the subgradients with respect to every node from the mixture (Sections 3 and 4). The second technical challenge is that most existing exact screening methods need to estimate an upper bound involving the dual optimum. This turns out to be a complicated *nonconvex* optimization problem for TGL. Thus, our second major technical contribution is to show that this highly nontrivial nonconvex optimization problem admits closed form solutions (Section 5). Experiments on both synthetic and real data sets demonstrate that the speedup gained by MLFre can be orders of magnitude (Section 6). Please see supplements for detailed proofs of the results in the main text.

**Notation**: Let $\|\cdot\|$ be the $\ell_2$ norm, $[p] = \{1, \ldots, p\}$ for a positive integer $p$, $G \subseteq [p]$, and $\bar{G} = [p] \backslash G$. For $\mathbf{u} \in \mathbb{R}^p$, let $u_i$ be its $i^{th}$ component. For $G \subseteq [p]$, we denote $\mathbf{u}_G = [\mathbf{u}]_G = \{\mathbf{v} : v_i = u_i \text{ if } i \in G, v_i = 0 \text{ otherwise}\}$ and $\mathcal{H}_G = \{\mathbf{u} \in \mathbb{R}^p : \mathbf{u}_{\bar{G}} = 0\}$. If $G_1, G_2 \subseteq [n]$ and $G_1 \subset G_2$, we emphasize that $G_2 \setminus G_1 \neq \emptyset$. For a set $\mathcal{C}$, let $\text{int}\,\mathcal{C}$, $\text{ri}\,\mathcal{C}$, $\text{bd}\,\mathcal{C}$, and $\text{rbd}\,\mathcal{C}$ be its interior, relative interior, boundary, and relative boundary, respectively [5]. If $\mathcal{C}$ is closed and convex, the projection operator is $\mathbf{P}_{\mathcal{C}}(\mathbf{z}) := \text{argmin}_{\mathbf{u} \in \mathcal{C}} \|\mathbf{z} - \mathbf{u}\|$, and its indicator function is $\mathbf{I}_{\mathcal{C}}(\cdot)$, which is 0 on $\mathcal{C}$ and $\infty$ elsewhere. Let $\Gamma_0(\mathbb{R}^p)$ be the class of proper closed convex functions on $\mathbb{R}^p$. For $f \in \Gamma_0(\mathbb{R}^p)$, let $\partial f$ be its subdifferential and $\text{dom}\, f := \{\mathbf{z} : f(\mathbf{z}) < \infty\}$. We denote by $\gamma_+ = \max(\gamma, 0)$.

## 2   Basics

We briefly review some basics of TGL. First, we introduce the so-called index tree.

**Definition 1.** [16] *For an index tree $T$ of depth $d$, we denote the node(s) of depth $i$ by $T_i = \{G_1^i, \ldots, G_{n_i}^i\}$, where $n_0 = 1$, $G_1^0 = [p]$, $G_j^i \subset [p]$, and $n_i \geq 1, \forall\, i \in [d]$. We assume that*
(i): $G_{j_1}^i \cap G_{j_2}^i = \emptyset, \forall\, i \in [d]$ and $j_1 \neq j_2$ *(different nodes of the same depth do not overlap).*
(ii): *If $G_j^i$ is a parent node of $G_\ell^{i+1}$, then $G_\ell^{i+1} \subset G_j^i$.*

When the tree structure is available (see supplement for an example), the TGL problem is

$$\min_{\beta} \; \tfrac{1}{2}\|\mathbf{y} - \mathbf{X}\beta\|^2 + \lambda \sum_{i=0}^{d} \sum_{j=1}^{n_i} w_j^i \|\beta_{G_j^i}\|, \tag{TGL}$$

where $\mathbf{y} \in \mathbb{R}^N$ is the response vector, $\mathbf{X} \in \mathbb{R}^{N \times p}$ is the data matrix, $\beta_{G_j^i}$ and $w_j^i$ are the coefficients vector and positive weight corresponding to node $G_j^i$, respectively, and $\lambda > 0$ is the regularization parameter. We derive the Lagrangian dual problem of TGL as follows.

**Theorem 2.** *For the TGL problem, let $\phi(\beta) = \sum_{i=0}^{d} \sum_{j=1}^{n_i} w_j^i \|\beta_{G_j^i}\|$. The following hold:*
(i): *Let $\phi_j^i(\beta) = \|\beta_{G_j^i}\|$ and $\mathcal{B}_j^i = \{\zeta \in \mathcal{H}_{G_j^i} : \|\zeta\| \leq w_j^i\}$. We can write $\partial\phi(0)$ as*

$$\partial\phi(0) = \sum_{i=0}^{d} \sum_{j=1}^{n_i} w_j^i \partial\phi_j^i(0) = \sum_{i=0}^{d} \sum_{j=1}^{n_i} \mathcal{B}_j^i. \tag{1}$$

(ii): *Let $\mathcal{F} = \{\theta : \mathbf{X}^T\theta \in \partial\phi(0)\}$. The Lagrangian dual of TGL is*

$$\sup_{\theta} \; \left\{ \tfrac{1}{2}\|\mathbf{y}\|^2 - \tfrac{1}{2}\|\tfrac{\mathbf{y}}{\lambda} - \theta\|^2 : \theta \in \mathcal{F} \right\}. \tag{2}$$

(iii): *Let $\beta^*(\lambda)$ and $\theta^*(\lambda)$ be the optimal solution of problems (TGL) and (2), respectively. Then,*

$$\mathbf{y} = \mathbf{X}\beta^*(\lambda) + \lambda\theta^*(\lambda), \tag{3}$$

$$\mathbf{X}^T\theta^*(\lambda) \in \sum_{i=0}^{d} \sum_{j=1}^{n_i} w_j^i \partial\phi_j^i(\beta^*(\lambda)). \tag{4}$$

The dual problem of TGL in (2) is equivalent to a projection problem, i.e., $\theta^*(\lambda) = \mathbf{P}_{\mathcal{F}}(\mathbf{y}/\lambda)$. This geometric property plays a fundamentally important role in developing MLFre (see Section 5).

## 3   Testing Dual Feasibility via Hierarchical Projection

Although the dual problem in (2) has nice geometric properties, it is challenging to determine the feasibility of a given $\theta$ due to the complex dual feasible set $\mathcal{F}$. An alternative approach is to test if $\mathbf{X}^T\theta = \mathbf{P}_{\partial\phi(0)}(\mathbf{X}^T\theta)$. Although $\partial\phi(0)$ is very complicated, we show that $\mathbf{P}_{\partial\phi(0)}(\cdot)$ admits a closed form solution by *hierarchically splitting* $\mathbf{P}_{\partial\phi(0)}(\cdot)$ into *a sum of projection operators with respect to a collection of simpler sets*. We first introduce some notations. For an index tree $T$, let

$$\mathcal{A}_j^i = \left\{ \sum_{t,k} \mathcal{B}_k^t : G_k^t \subseteq G_j^i \right\}, \forall\, i \in 0 \cup [d], j \in [n_i], \tag{5}$$

$$\mathcal{C}_j^i = \left\{ \sum_{t,k} \mathcal{B}_k^t : G_k^t \subset G_j^i \right\}, \forall\, i \in 0 \cup [d], j \in [n_i]. \tag{6}$$

For a node $G_j^i$, the set $\mathcal{A}_j^i$ is the sum of $\mathcal{B}_k^t$ corresponding to all its descendant nodes *and itself*, and the set $\mathcal{C}_j^i$ the sum *excluding itself*. Therefore, by the definitions of $\mathcal{A}_j^i$, $\mathcal{B}_j^i$, and $\mathcal{C}_j^i$, we have

$$\partial\phi(0) = \mathcal{A}_1^0, \qquad \mathcal{A}_j^i = \mathcal{B}_j^i + \mathcal{C}_j^i, \forall\, \text{non-leaf node } G_j^i, \qquad \mathcal{A}_j^i = \mathcal{B}_j^i, \forall\, \text{leaf node } G_j^i, \tag{7}$$

which implies that $\mathbf{P}_{\partial\phi(0)}(\cdot) = \mathbf{P}_{\mathcal{A}_1^0}(\cdot) = \mathbf{P}_{\mathcal{B}_1^0 + \mathcal{C}_1^0}(\cdot)$. This motivates the first pillar of this paper, i.e., Lemma 3, which splits $\mathbf{P}_{\mathcal{B}_1^0 + \mathcal{C}_1^0}(\cdot)$ into the sum of two projections onto $\mathcal{B}_1^0$ and $\mathcal{C}_1^0$, respectively.

**Lemma 3.** *Let $G \subseteq [p]$, $\mathcal{B} = \{\mathbf{u} \in \mathcal{H}_G : \|\mathbf{u}\| \le \gamma\}$ with $\gamma > 0$, $\mathcal{C} \subseteq \mathcal{H}_G$ a nonempty closed convex set, and $\mathbf{z}$ an arbitrary point in $\mathcal{H}_G$. Then, the following hold:*

(i): [2] $\mathbf{P}_{\mathcal{B}}(\mathbf{z}) = \min\{1, \gamma/\|\mathbf{z}\|\}\mathbf{z}$ *if* $\mathbf{z} \ne 0$. *Otherwise,* $\mathbf{P}_{\mathcal{B}}(\mathbf{z}) = 0$.

(ii): $\mathbf{I}_{\mathcal{B}+\mathcal{C}}(\mathbf{z}) = \mathbf{I}_{\mathcal{B}}(\mathbf{z} - \mathbf{P}_{\mathcal{C}}(\mathbf{z}))$, *i.e.,* $\mathbf{P}_{\mathcal{C}}(\mathbf{z}) \in \arg\min_{\mathbf{u}\in\mathcal{C}} \mathbf{I}_{\mathcal{B}}(\mathbf{z} - \mathbf{u})$.

(iii): $\mathbf{P}_{\mathcal{B}+\mathcal{C}}(\mathbf{z}) = \mathbf{P}_{\mathcal{C}}(\mathbf{z}) + \mathbf{P}_{\mathcal{B}}(\mathbf{z} - \mathbf{P}_{\mathcal{C}}(\mathbf{z}))$.

By part (iii) of Lemma 3, we can split $\mathbf{P}_{\mathcal{A}_1^0}(\mathbf{X}^T\theta)$ in the following form:

$$\mathbf{P}_{\mathcal{A}_1^0}(\mathbf{X}^T\theta) = \mathbf{P}_{\mathcal{C}_1^0}(\mathbf{X}^T\theta) + \mathbf{P}_{\mathcal{B}_1^0}(\mathbf{X}^T\theta - \mathbf{P}_{\mathcal{C}_1^0}(\mathbf{X}^T\theta)). \tag{8}$$

As $\mathbf{P}_{\mathcal{B}_1^0}(\cdot)$ admits a closed form solution by part (i) of Lemma 3, we can compute $\mathbf{P}_{\mathcal{A}_1^0}(\mathbf{X}^T\theta)$ if we have $\mathbf{P}_{\mathcal{C}_1^0}(\mathbf{X}^T\theta)$ computed. By Eq. (5) and Eq. (6), for a non-leaf node $G_j^i$, we note that

$$\mathcal{C}_j^i = \sum_{k \in \mathcal{I}_c(G_j^i)} \mathcal{A}_k^{i+1}, \text{ where } \mathcal{I}_c(G_j^i) = \{k : G_k^{i+1} \subset G_j^i\}. \tag{9}$$

Inspired by (9), we have the following result.

**Lemma 4.** *Let $\{G_\ell \subset [p]\}_\ell$ be a set of nonoverlapping index sets, $\{\mathcal{C}_\ell \subseteq \mathcal{H}_{G_\ell}\}_\ell$ be a set of nonempty closed convex sets, and $\mathcal{C} = \sum_\ell \mathcal{C}_\ell$. Then, $\mathbf{P}_{\mathcal{C}}(\mathbf{z}) = \sum_\ell \mathbf{P}_{\mathcal{C}_\ell}(\mathbf{z}_{G_\ell})$ for $\mathbf{z} \in \mathbb{R}^p$.*

**Remark 1.** *For Lemma 4, if all $\mathcal{C}_\ell$ are balls centered at $0$, then $\mathbf{P}_{\mathcal{C}}(\mathbf{z})$ admits a closed form solution.*

By Lemma 4 and Eq. (9), we can further splits $\mathbf{P}_{\mathcal{C}_1^0}(\mathbf{X}^T\theta)$ in Eq. (8) in the following form.

$$\mathbf{P}_{\mathcal{C}_1^0}(\mathbf{X}^T\theta) = \sum_{k \in \mathcal{I}_c(G_1^0)} \mathbf{P}_{\mathcal{A}_k^1}([\mathbf{X}^T\theta]_{G_k^1}), \text{ where } \mathcal{I}_c(G_1^0) = \{k : G_k^1 \subset G_1^0\}. \tag{10}$$

Consider the right hand side of Eq. (10). If $G_k^1$ is a leaf node, Eq. (7) implies that $\mathcal{A}_k^1 = \mathcal{B}_k^1$ and thus $\mathbf{P}_{\mathcal{A}_k^1}(\cdot)$ admits a closed form solution by part (i) of Lemma 3. Otherwise, we continue to split $\mathbf{P}_{\mathcal{A}_k^1}(\cdot)$ by Lemmas (3) and (4). This procedure will always terminate as we reach the leaf nodes [see the last equality in Eq. (7)]. Therefore, by a repeated application of Lemmas (3) and (4), the following algorithm computes the closed form solution of $\mathbf{P}_{\mathcal{A}_1^0}(\cdot)$.

---

**Algorithm 1** Hierarchical Projection: $\mathbf{P}_{\mathcal{A}_1^0}(\cdot)$.

---

**Input:** $\mathbf{z} \in \mathbb{R}^p$, the index tree $T$ as in Definition 1, and positive weights $w_j^i$ for all nodes $G_j^i$ in $T$.
**Output:** $\mathbf{u}^0 = \mathbf{P}_{\mathcal{A}_1^0}(\mathbf{z})$, $\mathbf{v}^i$ for $\forall\, i \in 0 \cup [d]$.

  1: Set $\mathbf{u}^i \leftarrow 0 \in \mathbb{R}^p$, $\forall\, i \in 0 \cup [d+1]$, $\mathbf{v}^i \leftarrow 0 \in \mathbb{R}^p$, $\forall\, i \in 0 \cup [d]$.
  2: **for** $i = d$ to $0$ **do**                                         /\*hierarchical projection\*/
  3:     **for** $j = 1$ to $n_i$ **do**

$$\mathbf{v}_{G_j^i}^i = \mathbf{P}_{\mathcal{B}_j^i}(\mathbf{z}_{G_j^i} - \mathbf{u}_{G_j^i}^{i+1}), \tag{11}$$

$$\mathbf{u}_{G_j^i}^i \leftarrow \mathbf{u}_{G_j^i}^{i+1} + \mathbf{v}_{G_j^i}^i. \tag{12}$$

  5:     **end for**
  6: **end for**

---

The time complexity of Algorithm 1 is similar to that of solving its proximal operator [16], i.e., $O(\sum_{i=0}^{d}\sum_{j=1}^{n_i}|G_j^i|)$, where $|G_j^i|$ is the number of features contained in the node $G_j^i$. As $\sum_{j=1}^{n_i}|G_j^i| \leq p$ by Definition 1, the time complexity of Algorithm 1 is $O(pd)$, and thus $O(p\log p)$ for a balanced tree, where $d = O(\log p)$. The next result shows that $\mathbf{u}^0$ returned by Algorithm 1 is the projection of $\mathbf{z}$ onto $\mathcal{A}_1^0$. Indeed, we have more general results as follows.

**Theorem 5.** *For Algorithm* 1, *the following hold:*

(i): $\mathbf{u}_{G_j^i}^i = \mathbf{P}_{\mathcal{A}_j^i}\left(\mathbf{z}_{G_j^i}\right)$, $\forall\, i \in 0 \cup [d]$, $j \in [n_i]$.

(ii): $\mathbf{u}_{G_j^i}^{i+1} = \mathbf{P}_{\mathcal{C}_j^i}\left(\mathbf{z}_{G_j^i}\right)$, *for any non-leaf node* $G_j^i$.

## 4  MLFre Inspired by the KKT Conditions and Hierarchical Projection

In this section, we motivate MLFre via the KKT condition in Eq. (4) and the hierarchical projection in Algorithm 1. Note that for any node $G_j^i$, we have

$$w_j^i \partial \phi_j^i(\beta^*(\lambda)) = \begin{cases} \{\zeta \in \mathcal{H}_{G_j^i} : \|\zeta\| \leq w_j^i\}, & \text{if } [\beta^*(\lambda)]_{G_j^i} = 0, \\ w_j^i[\beta^*(\lambda)]_{G_j^i}/\|[\beta^*(\lambda)]_{G_j^i}\|, & \text{otherwise.} \end{cases} \tag{13}$$

Moreover, the KKT condition in Eq. (4) implies that

$$\exists\, \{\xi_j^i \in w_j^i \partial \phi_j^i(\beta^*(\lambda)) : \forall\, i \in 0 \cup [d], j \in [n_i]\} \text{ such that } \mathbf{X}^T\theta^*(\lambda) = \sum_{i=0}^{d}\sum_{j=1}^{n_i} \xi_j^i. \tag{14}$$

Thus, if $\|\xi_j^i\| < w_j^i$, we can see that $[\beta^*(\lambda)]_{G_j^i} = 0$. However, we do not have direct access to $\xi_j^i$ even if $\theta^*(\lambda)$ is known, because $\mathbf{X}^T\theta^*(\lambda)$ is a mixture (sum) of all $\xi_j^i$ as shown in Eq. (14). Indeed, Algorithm 1 turns out to be much more useful than testing the feasibility of a given $\theta$: it is able to split all $\xi_j^i \in w_j^i \partial \phi_j^i(\beta^*(\lambda))$ from $\mathbf{X}^T\theta^*(\lambda)$. This will serve as a cornerstone in developing MLFre. Theorem 6 rigorously shows this property of Algorithm 1.

**Theorem 6.** *Let* $\mathbf{v}^i$, $i \in 0 \cup [d]$ *be the output of Algorithm* 1 *with input* $\mathbf{X}^T\theta^*(\lambda)$, *and* $\{\xi_j^i : i \in 0 \cup [d], j \in [n_i]\}$ *be the set of vectors that satisfy Eq.* (14). *Then, the following hold.*

(i) *If* $[\beta^*(\lambda)]_{G_j^i} = 0$, *and* $[\beta^*(\lambda)]_{G_r^l} \neq 0$ *for all* $G_r^l \supset G_j^i$, *then*
$$\mathbf{P}_{\mathcal{A}_j^i}\left([\mathbf{X}^T\theta^*(\lambda)]_{G_j^i}\right) = \sum_{\{(k,t):G_k^t \subseteq G_j^i\}} \xi_k^t.$$

(ii) *If* $G_j^i$ *is a non-leaf node, and* $[\beta^*(\lambda)]_{G_j^i} \neq 0$, *then*
$$\mathbf{P}_{\mathcal{C}_j^i}\left([\mathbf{X}^T\theta^*(\lambda)]_{G_j^i}\right) = \sum_{\{(k,t):G_k^t \subset G_j^i\}} \xi_k^t.$$

(iii) $\mathbf{v}_{G_j^i}^i \in w_j^i \partial \phi_j^i(\beta^*(\lambda))$, $\forall\, i \in 0 \cup [d]$, $j \in [n_i]$.

Combining Eq. (13) and part (iii) of Theorem 6, we can see that

$$\|\mathbf{v}_{G_j^i}^i\| < w_j^i \Rightarrow [\beta^*(\lambda)]_{G_j^i} = 0. \tag{15}$$

By plugging Eq. (11) and part (ii) of Theorem 5 into (15), we have $[\beta^*(\lambda)]_{G_j^i} = 0$ if

(a): $\left\|\mathbf{P}_{\mathcal{B}_j^i}\left([\mathbf{X}^T\theta^*(\lambda)]_{G_j^i} - \mathbf{P}_{\mathcal{C}_j^i}\left([\mathbf{X}^T\theta^*(\lambda)]_{G_j^i}\right)\right)\right\| < w_j^i$, if $G_j^i$ is a non-leaf node,　(R1)

(b): $\left\|\mathbf{P}_{\mathcal{B}_j^i}\left([\mathbf{X}^T\theta^*(\lambda)]_{G_j^i}\right)\right\| < w_j^i$,　　　　　　　　　　if $G_j^i$ is a leaf node.　(R2)

Moreover, the definition of $\mathbf{P}_{\mathcal{B}_j^i}$ implies that we can simplify (R1) and (R2) to the following form:

$$\left\|[\mathbf{X}^T\theta^*(\lambda)]_{G_j^i} - \mathbf{P}_{\mathcal{C}_j^i}\left([\mathbf{X}^T\theta^*(\lambda)]_{G_j^i}\right)\right\| < w_j^i \Rightarrow [\beta^*(\lambda)]_{G_j^i} = 0, \text{if } G_j^i \text{ is a non-leaf node,　(R1')}$$

$$\left\|[\mathbf{X}^T\theta^*(\lambda)]_{G_j^i}\right\| < w_j^i \Rightarrow [\beta^*(\lambda)]_{G_j^i} = 0,\qquad\qquad\qquad \text{if } G_j^i \text{ is a leaf node.　(R2')}$$

However, (R1') and (R2') are not applicable to detect inactive nodes as they involve $\theta^*(\lambda)$. Inspired by SAFE [8], we first estimate a set $\Theta$ containing $\theta^*(\lambda)$. Let $[\mathbf{X}^T\Theta]_{G_j^i} = \{[\mathbf{X}^T\theta]_{G_j^i} : \theta \in \Theta\}$ and

$$\mathbf{S}_j^i(\mathbf{z}) = \mathbf{z}_{G_j^i} - \mathbf{P}_{\mathcal{C}_j^i}\left(\mathbf{z}_{G_j^i}\right). \tag{16}$$

Then, we can relax (R1') and (R2') as

$$\sup_\zeta \left\{ \|\mathbf{S}_j^i(\zeta)\| : \zeta_{G_j^i} \in \Xi_j^i \supseteq [\mathbf{X}^T\Theta]_{G_j^i} \right\} < w_j^i \Rightarrow [\beta^*(\lambda)]_{G_j^i} = 0, \text{if } G_j^i \text{ is a non-leaf node}, \quad (\text{R1}^*)$$

$$\sup_\zeta \left\{ \left\|\zeta_{G_j^i}\right\| : \zeta_{G_j^i} \in [\mathbf{X}^T\Theta]_{G_j^i} \right\} < w_j^i \Rightarrow [\beta^*(\lambda)]_{G_j^i} = 0, \qquad \text{if } G_j^i \text{ is a leaf node}. \quad (\text{R2}^*)$$

In view of (R1$^*$) and (R2$^*$), we sketch the procedure to develop MLFre in the following three steps.
**Step 1** We estimate a set $\Theta$ that contains $\theta^*(\lambda)$.
**Step 2** We solve for the supreme values in (R1$^*$) and (R2$^*$), respectively.
**Step 3** We develop MLFre by plugging the supreme values obtained in **Step 2** to (R1$^*$) and (R2$^*$).

### 4.1   The Effective Interval of the Regularization Parameter $\lambda$

The geometric property of the dual problem in (2), i.e., $\theta^*(\lambda) = \mathbf{P}_\mathcal{F}(\mathbf{y}/\lambda)$, implies that $\theta^*(\lambda) = \mathbf{y}/\lambda$ if $\mathbf{y}/\lambda \in \mathcal{F}$. Moreover, (R1) for the root node $G_1^0$ leads to $\beta^*(\lambda) = 0$ if $\mathbf{y}/\lambda$ is an interior point of $\mathcal{F}$. Indeed, the following theorem presents stronger results.

**Theorem 7.** *For TGL, let* $\lambda_{\max} = \max\{\lambda : \mathbf{y}/\lambda \in \mathcal{F}\}$ *and* $\mathbf{S}_1^0(\cdot)$ *be defined by Eq. (16). Then,*

(i): $\lambda_{\max} = \{\lambda : \|\mathbf{S}_1^0(\mathbf{X}^T\mathbf{y}/\lambda)\| = w_1^0\}$.

(ii): $\frac{\mathbf{y}}{\lambda} \in \mathcal{F} \Leftrightarrow \lambda \geq \lambda_{\max} \Leftrightarrow \theta^*(\lambda) = \frac{\mathbf{y}}{\lambda} \Leftrightarrow \beta^*(\lambda) = 0$.

For more discussions on $\lambda_{\max}$, please refer to Section H in the supplements.

## 5   The Proposed Multi-Layer Feature Reduction Method for TGL

We follow the three steps in Section 4 to develop MLFre. Specifically, we first present an accurate estimation of the dual optimum in Section 5.1, then we solve for the supreme values in (R1$^*$) and (R2$^*$) in Section 5.2, and finally we present the proposed MLFre in Section 5.3.

### 5.1   Estimation of the Dual Optimum

We estimate the dual optimum by the geometric properties of projection operators [recall that $\theta^*(\lambda) = \mathbf{P}_\mathcal{F}(\mathbf{y}/\lambda)$]. We first introduce a useful tool to characterize the projection operators.

> **Definition 8.** [2] *For a closed convex set $\mathcal{C}$ and a point $\mathbf{z}_0 \in \mathcal{C}$, the normal cone to $\mathcal{C}$ at $\mathbf{z}_0$ is*
> $$N_\mathcal{C}(\mathbf{z}_0) = \{\zeta : \langle\zeta, \mathbf{z} - \mathbf{z}_0\rangle \leq 0, \, \forall \, \mathbf{z} \in \mathcal{C}\}.$$

Theorem 7 implies that $\theta^*(\lambda)$ is known with $\lambda \geq \lambda_{\max}$. Thus, we can estimate $\theta^*(\lambda)$ in terms of a known $\theta^*(\lambda_0)$. This leads to Theorem 9 that bounds the dual optimum by a small ball.

**Theorem 9.** *For TGL, suppose that $\theta^*(\lambda_0)$ is known with $\lambda_0 \leq \lambda_{\max}$. For $\lambda \in (0, \lambda_0)$, we define*

$$\mathbf{n}(\lambda_0) = \begin{cases} \frac{\mathbf{y}}{\lambda_0} - \theta^*(\lambda_0), & \text{if } \lambda_0 < \lambda_{\max}, \\ \mathbf{X}\mathbf{S}_1^0\left(\mathbf{X}^T\frac{\mathbf{y}}{\lambda_{\max}}\right), & \text{if } \lambda_0 = \lambda_{\max}, \end{cases}$$

$$\mathbf{r}(\lambda, \lambda_0) = \frac{\mathbf{y}}{\lambda} - \theta^*(\lambda_0),$$

$$\mathbf{r}^\perp(\lambda, \lambda_0) = \mathbf{r}(\lambda, \lambda_0) - \frac{\langle\mathbf{r}(\lambda,\lambda_0),\mathbf{n}(\lambda_0)\rangle}{\|\mathbf{n}(\lambda_0)\|^2}\mathbf{n}(\lambda_0).$$

*Then, the following hold:*

(i): $\mathbf{n}(\lambda_0) \in N_\mathcal{F}(\theta^*(\lambda_0))$.
(ii): $\|\theta^*(\lambda) - (\theta^*(\lambda_0) + \frac{1}{2}\mathbf{r}^\perp(\lambda, \lambda_0))\| \leq \frac{1}{2}\|\mathbf{r}^\perp(\lambda, \lambda_0)\|$.

Theorem 9 indicates that $\theta^*(\lambda)$ lies inside the ball of radius $\frac{1}{2}\|\mathbf{r}^\perp(\lambda, \lambda_0)\|$ centered at
$$\mathbf{o}(\lambda, \lambda_0) = \theta^*(\lambda_0) + \frac{1}{2}\mathbf{r}^\perp(\lambda, \lambda_0).$$

### 5.2   Solving the Nonconvex Optimization Problems in (R1$^*$) and (R2$^*$)

We solve for the supreme values in (R1$^*$) and (R2$^*$). For notational convenience, let

$$\Theta = \{\theta : \|\theta - \mathbf{o}(\lambda, \lambda_0)\| \leq \tfrac{1}{2}\|\mathbf{r}^\perp(\lambda, \lambda_0)\|\}, \tag{17}$$

$$\Xi_j^i = \{\zeta : \zeta \in \mathcal{H}_{G_j^i}, \|\zeta - [\mathbf{X}^T\mathbf{o}(\lambda, \lambda_0)]_{G_j^i}\| \leq \tfrac{1}{2}\|\mathbf{r}^\perp(\lambda, \lambda_0)\|\|\mathbf{X}_{G_j^i}\|_2\}. \tag{18}$$

Theorem 9 implies that $\theta^*(\lambda) \in \Theta$, and thus $[\mathbf{X}^T\Theta]_{G_j^i} \subseteq \Xi_j^i$ for all non-leaf nodes $G_j^i$. To develop MLFre by (R1$^*$) and (R2$^*$), we need to solve the following optimization problems:

$$s_j^i(\lambda, \lambda_0) = \sup_\zeta\{\|\mathbf{S}_j^i(\zeta)\| : \zeta \in \Xi_j^i\}, \quad \text{if } G_j^i \text{ is a non-leaf node}, \tag{19}$$

$$s_j^i(\lambda, \lambda_0) = \sup_\zeta\{\|\zeta\| : \zeta \in \Xi_j^i\}, \qquad \text{if } G_j^i \text{ is a leaf node}. \tag{20}$$

Before we solve problems (19) and (20), we first introduce some notations.

**Definition 10.** *For a non-leaf node $G_j^i$ of an index tree $T$, let $\mathcal{I}_c(G_j^i) = \{k : G_k^{i+1} \subset G_j^i\}$. If $G_j^i \setminus \cup_{k \in \mathcal{I}_c(G_j^i)} G_k^{i+1} \neq \emptyset$, we define a virtual child node of $G_j^i$ by $G_{j'}^{i+1} = G_j^i \setminus \cup_{k \in \mathcal{I}_c(G_j^i)} G_k^{i+1}$ for $j' \in \{n_{i+1} + 1, n_{i+1} + 2, \ldots, n_{i+1} + n'_{i+1}\}$, where $n'_{i+1}$ is the number of virtual nodes of depth $i + 1$. We set the weights $w_{j'}^i = 0$ for all virtual nodes $G_{j'}^i$.*

Another useful concept is the so-called unique path between the nodes in the tree.

**Lemma 11.** *[16] For any non-root node $G_j^i$, we can find a unique path from $G_j^i$ to the root $G_1^0$. Let the nodes on this path be $G_{r_l}^l$, where $l \in 0 \cup [i]$, $r_0 = 1$, and $r_i = j$. Then, the following hold:*

$$G_j^i \subset G_{r_l}^l, \quad \forall l \in 0 \cup [i-1]. \tag{21}$$

$$G_j^i \cap G_r^l = \emptyset, \quad \forall r \neq r_l, l \in [i-1], r \in [n_i]. \tag{22}$$

**Solving Problem (19)** We consider the following equivalent problem of (19).

$$\tfrac{1}{2}(s_j^i(\lambda, \lambda_0))^2 = \sup_\zeta \{\tfrac{1}{2}\|\mathbf{S}_j^i(\zeta)\|^2 : \zeta \in \Xi_j^i\}, \quad \text{if } G_j^i \text{ is a non-leaf node.} \tag{23}$$

Although both the objective function and feasible set of problem (23) are convex, it is nonconvex as we need to find the supreme value. We derive the closed form solutions of (19) and (23) as follows.

**Theorem 12.** *Let $\mathbf{c} = [\mathbf{X}^T \mathbf{o}(\lambda, \lambda_0)]_{G_j^i}$, $\gamma = \tfrac{1}{2}\|\mathbf{r}^\perp(\lambda, \lambda_0)\|\|\mathbf{X}_{G_j^i}\|_2$, and $\mathbf{v}^i$, $i \in 0 \cup [d]$ be the output of Algorithm 1 with input $\mathbf{X}^T \mathbf{o}(\lambda, \lambda_0)$.*

*(i): Suppose that $\mathbf{c} \notin \mathcal{C}_j^i$. Then, $s_j^i(\lambda, \lambda_0) = \|\mathbf{v}_{G_j^i}^i\| + \gamma$.*

*(ii): Suppose that node $G_j^i$ has a virtual child node. Then, for any $\mathbf{c} \in \mathcal{C}_j^i$, $s_j^i(\lambda, \lambda_0) = \gamma$.*

*(iii): Suppose that node $G_j^i$ has no virtual child node. Then, the following hold.*

  *(iii.a): If $\mathbf{c} \in \text{rbd } \mathcal{C}_j^i$, then $s_j^i(\lambda, \lambda_0) = \gamma$.*

  *(iii.b): If $\mathbf{c} \in \text{ri } \mathcal{C}_j^i$, then, for any node $G_k^t \subset G_j^i$, where $t \in \{i + 1, \ldots, d\}$ and $k \in [n_t + n'_t]$, let the nodes on the path from $G_k^t$ to $G_j^i$ be $G_{r_l}^l$, where $l = i, \ldots, t$, $r_i = j$, and $r_t = k$, and*

$$\Gamma(G_{r_{i+1}}^{i+1}, G_k^t) = \sum_{l=i+1}^t \left( w_{r_l}^l - \|\mathbf{v}_{G_{r_l}^l}^l\| \right). \tag{24}$$

  *Then, $s_j^i(\lambda, \lambda_0) = \left( \gamma - \min_{\{(k,t):G_k^t \subset G_j^i\}} \Gamma(G_{r_{i+1}}^{i+1}, G_k^t) \right)_+$.*

**Solving Problem (20)** We can solve problem (20) by the Cauchy-Schwarz inequality.

**Theorem 13.** *For problem (20), we have $s_j^i(\lambda, \lambda_0) = \|[\mathbf{X}^T \mathbf{o}(\lambda, \lambda_0)]_{G_j^i}\| + \tfrac{1}{2}\|\mathbf{r}^\perp(\lambda, \lambda_0)\|\|\mathbf{X}_{G_j^i}\|_2$.*

### 5.3 The Multi-Layer Screening Rule

In real-world applications, the optimal parameter values are usually unknown. Commonly used approaches to determine an appropriate parameter value, such as cross validation and stability selection, solve TGL many times along a grid of parameter values. This process can be very time consuming. Motivated by this challenge, we present MLFre in the following theorem by plugging the supreme values found by Theorems 12 and 13 into (R1$^*$) and (R2$^*$), respectively.

**Theorem 14.** *For the TGL problem, suppose that we are given a sequence of parameter values $\lambda_{\max} = \lambda_0 > \lambda_1 > \cdots > \lambda_\mathcal{K}$. For each integer $k = 0, \ldots, \mathcal{K} - 1$, we compute $\theta^*(\lambda_k)$ from a given $\beta^*(\lambda_k)$ via Eq. (3). Then, for $i = 1, \ldots, d$, MLFre takes the form of*

$$s_j^i(\lambda_{k+1}, \lambda_k) < w_j^i \Rightarrow [\beta^*(\lambda)]_{G_j^i} = 0, \forall j \in [n_i]. \tag{MLFre}$$

**Remark 2.** *We apply MLFre to identify inactive nodes hierarchically in a top-down fashion. Note that, we do not need to apply MLFre to node $G_j^i$ if one of its ancestor nodes passes the rule.*

**Remark 3.** *To simplify notations, we consider TGL with a single tree, in the proof. However, all major results are directly applicable to TGL with multiple trees, as they are independent from each other. We note that, many sparse models, such as Lasso, group Lasso, and sparse group Lasso, are special cases of TGL with multiple trees.*

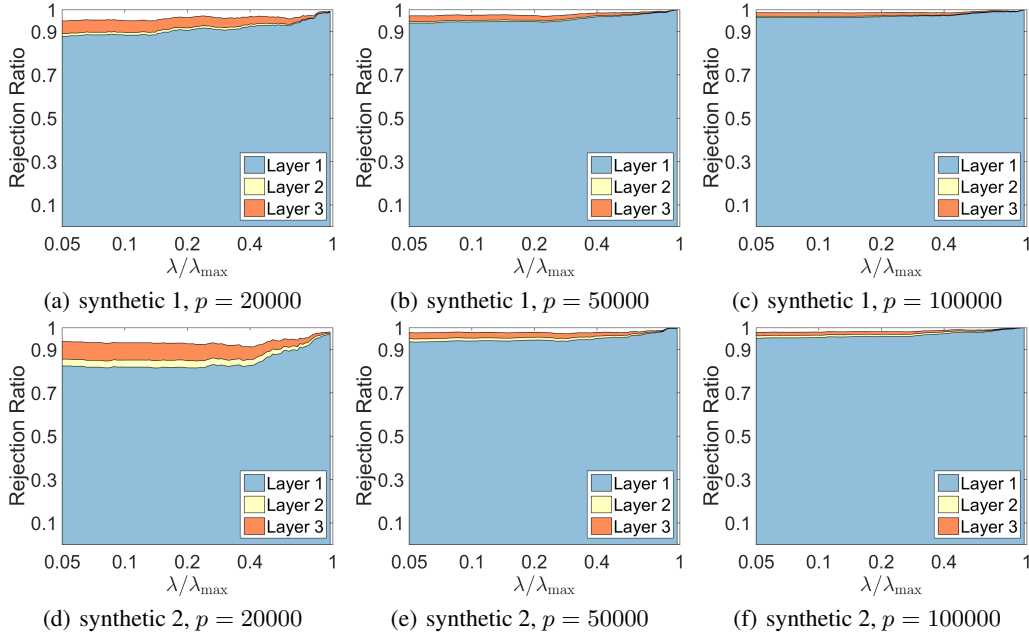

Figure 1: Rejection ratios of MLFre on two synthetic data sets with different feature dimensions.

## 6 Experiments

We evaluate MLFre on both synthetic and real data sets by two measurements. The first measure is the *rejection ratios* of MLFre for each level of the tree. Let $p_0$ be the number of zero coefficients in the solution vector and $\mathcal{G}^i$ be the index set of the inactive nodes with depth $i$ identified by MLFre. The rejection ratio of the $i^{th}$ layer of MLFre is defined by $r_i = \frac{\sum_{k \in \mathcal{G}^i} |G_k^i|}{p_0}$, where $|G_k^i|$ is the number of features contained in node $G_k^i$. The second measure is *speedup*, namely, the ratio of the running time of the solver without screening to the running time of solver with MLFre.

For each data set, we run the solver combined with MLFre along a sequence of 100 parameter values equally spaced on the logarithmic scale of $\lambda/\lambda_{\max}$ from 1.0 to 0.05. The solver for TGL is from the SLEP package [15]. It also provides an efficient routine to compute $\lambda_{\max}$.

### 6.1 Simulation Studies

We perform experiments on two synthetic data sets, named synthetic 1 and synthetic 2, which are commonly used in the literature [21, 31]. The true model is $\mathbf{y} = \mathbf{X}\beta^* + 0.01\epsilon$, $\epsilon \sim N(0, 1)$. For each of the data set, we fix $N = 250$ and select $p = 20000, 50000, 100000$. We create a tree with height 4, i.e., $d = 3$. The average sizes of the nodes with depth 1, 2 and 3 are 50, 10, and 1, respectively. Thus, if $p = 100000$, we have roughly $n_1 = 2000$, $n_2 = 10000$, and $n_3 = 100000$. For synthetic 1, the entries of the data matrix $\mathbf{X}$ are i.i.d. standard Gaussian with zero pair-wise correlation, i.e., corr $(\mathbf{x}_i, \mathbf{x}_j) = 0$ for the $i^{th}$ and $j^{th}$ columns of $\mathbf{X}$ with $i \neq j$. For synthetic 2, the entries of $\mathbf{X}$ are drawn from standard Gaussian with pair-wise correlation corr $(\mathbf{x}_i, \mathbf{x}_j) =$

Table 1: Running time (in seconds) for solving TGL along a sequence of 100 tuning parameter values of $\lambda$ equally spaced on the logarithmic scale of $\lambda/\lambda_{\max}$ from 1.0 to 0.05 by (a): the solver [15] without screening (see the third column); (b): the solver with MLFre (see the fifth column).

| Dataset | $p$ | solver | MLFre | MLFre+solver | speedup |
|---|---|---|---|---|---|
| synthetic 1 | 20000 | 483.96 | 1.03 | 30.17 | **16.04** |
| | 50000 | 1175.91 | 2.95 | 39.49 | **29.78** |
| | 100000 | 2391.43 | 6.57 | 58.91 | **40.60** |
| synthetic 2 | 20000 | 470.54 | 1.19 | 37.87 | **12.43** |
| | 50000 | 1122.30 | 3.13 | 43.97 | **25.53** |
| | 100000 | 2244.06 | 6.18 | 60.96 | **36.81** |
| ADNI+GMV | 406262 | 20911.92 | 81.14 | 492.08 | **42.50** |
| ADNI+WMV | 406262 | 21855.03 | 80.83 | 556.19 | **39.29** |
| ADNI+WBV | 406262 | 20812.06 | 82.10 | 564.36 | **36.88** |

$0.5^{|i-j|}$. To construct $\beta^*$, we first randomly select 50% of the nodes with depth 1, and then randomly select 20% of the children nodes (with depth 2) of the remaining nodes with depth 1. The components of $\beta^*$ corresponding to the remaining nodes are populated from a standard Gaussian, and the remaining ones are set to zero.

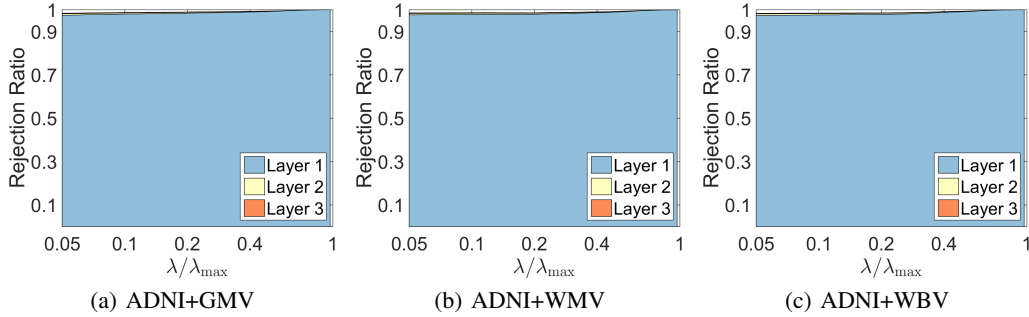

Figure 2: Rejection ratios of MLFre on ADNI data set with grey matter volume (GMV), white mater volume (WMV), and whole brain volume (WBV) as response vectors, respectively.

Fig. 1 shows the rejection ratios of all three layers of MLFre. We can see that MLFre identifies almost all of the inactive nodes, i.e., $\sum_{i=1}^{3} r_i \geq 90\%$, and the first layer contributes the most. Moreover, Fig. 1 also indicates that, as the feature dimension (and the number of nodes in each level) increases, MLFre identifies more inactive nodes, i.e., $\sum_{i=1}^{3} r_i \approx 100\%$. Thus, we can expect a more significant capability of MLFre in identifying inactive nodes on data sets with higher dimensions.

Table 1 shows the running time of the solver with and without MLFre. We can observe significant speedups gained by MLFre, which are up to 40 times. Take synthetic 1 with $p = 100000$ for example. The solver without MLFre takes about 40 minutes to solve TGL at 100 parameter values. Combined with MLFre, the solver only needs less than one minute for the same task. Table 1 also shows that the computational cost of MLFre is very low—that is negligible compared to that of the solver without MLFre. Moreover, as MLFre identifies more inactive nodes with increasing feature dimensions, Table 1 shows that the speedup gained by MLFre becomes more significant as well.

### 6.2 Experiments on ADNI data set

We perform experiments on the Alzheimers Disease Neuroimaging Initiative (ADNI) data set (http://adni.loni.usc.edu/). The data set consists of 747 patients with 406262 single nucleotide polymorphisms (SNPs). We create the index tree such that $n_1 = 4567$, $n_2 = 89332$, and $n_3 = 406262$. Fig. 2 presents the rejection ratios of MLFre on the ADNI data set with grey matter volume (GMV), white matter volume (WMV), and whole brain volume (WBV) as response, respectively. We can see that MLFre identifies almost all inactive nodes, i.e., $\sum_{i=1}^{3} r_i \approx 100\%$. As a result, we observe significant speedups gained by MLFre—that are about 40 times—from Table 1. Specifically, with GMV as response, the solver without MLFre takes about six hours to solve TGL at 100 parameter values. However, combined with MLFre, the solver only needs about eight minutes for the same task. Moreover, Table 1 also indicates that the computational cost of MLFre is very low—that is negligible compared to that of the solver without MLFre.

## 7 Conclusion

In this paper, we propose a novel multi-layer feature reduction (MLFre) method for TGL. Our major technical contributions lie in two folds. The first is the novel hierarchical projection algorithm that is able to exactly and efficiently recover the subgradients of the tree-guided regularizer with respect to each node from their mixture. The second is that we show a highly nontrivial nonconvex problem admits a closed form solution. To the best of our knowledge, MLFre is the first screening method that is applicable to TGL. An appealing feature of MLFre is that it is exact in the sense that the identified inactive nodes are guaranteed to be absent from the sparse representations. Experiments on both synthetic and real data sets demonstrate that MLFre is very effective in identifying inactive nodes, leading to substantial savings in computational cost and memory usage without sacrificing accuracy. Moreover, the capability of MLFre in identifying inactive nodes on higher dimensional data sets is more significant. We plan to generalize MLFre to more general and complicated sparse models, e.g., over-lapping group Lasso with logistic loss. In addition, we plan to apply MLFre to other applications, e.g., brain image analysis [10, 18] and natural language processing [27, 28].

## Acknowledgments

This work is supported in part by research grants from NIH (R01 LM010730, U54 EB020403) and NSF (IIS- 0953662, III-1539991, III-1539722).

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
