[Reviews · NeurIPS 2015]

Submitted by Assigned_Reviewer_1

This paper presented a Multi-Layer Feature reduction method (MLFre) to quickly identify the inactive nodes hierarchically in a top-down fashion. The proposed method has been studied theoretically. Experimental results show much improved efficiency than competing methods.

I just have two questions. In real-applications, how to build such a tree structure in the paper? Second, how to guarantee this tree structure is really helpful to improve the performance?
Summary: This paper presented a Multi-Layer Feature reduction method (MLFre) to quickly identify the inactive nodes hierarchically in a top-down fashion. The proposed method has been studied theoretically. Experimental results show much improved efficiency than competing methods.

Submitted by Assigned_Reviewer_2

The authors present a node screening method for tree-based regularization which identifies inactive nodes (i.e. nodes the features of which have all zero coefficients) in a top down manner given the tree that defines the groups of variables. As usual the idea is to work with the sub-gradient of the regularizer, if the subgradient with respect to a node is in the strict set of subgradient at 0, then this node is set to zero. The problem in tree based group lasso is that it is hard to access the subgradient of each node, since we only have access at the mixture of the subgradients with respect to all nodes. The authors provide a method to project the mixture of subgradients at each node and examine if the subgradient falls in the region which give us sparsity, if yes then by eliminating that corresponding node we remove those inactive features.

The proposed method makes the use of tree structured lasso on datasets with tens of thousands, or even hunderds of thousands of attributes.

The authors start by deriving the dual of the tree group lasso (which is equivalent to a projection problem on the feasible set), with dual variable $\vect \theta$ and feasible set F={$X^T \theta$ in the subdifferential at zero of the tree norm}.

They start with the observation that some $\theta$ will be a part of the feasible set if $X^T\theta$ is on the feasible set and thus its projection on the feasible set is $X^\theta$ and they provide an hierarchical algorithm that given a vector $z$ projects it on the feasible set. The algorithm works by hierarchically decomposing the projection on the subdifferential at zero to a projection on the sum of two sets: the bounded norm ball set B and the sum of the sets of the bounded norm balls of all of each children nodes. The sizes of the norm balls are given by the importance, w^i_j, of the respective nodes, G^i_j.

Using the hierarchical projection algorithm they can decompose $X^T\theta^*$ at the optimal solution as a sum of vectors $\ksi_j^i$ where each $\ksi_j^i$ is in the sudifferential set of tree norm with respect to the G^i_j node of the tree at the optimal solution of the primal $\beta^*$. Thus if one knows $X^T\theta^*$ then one can compute the inactive nodes, G_j^i, by looking at the $\ksi^i_j$ that are within the respective bounded radius ball. However since $\theta^*$ is not known the authors define a set that contains the optimal solution and solve for an upper bounding problem, providing along the way a close-form solution for the non-convex supremum problem.

The screening rule takes as input a sequence of hyperparameters $\lambda$ and for each sub-sequent pair outputs the corresponding inactive nodes of the tree, i.e. thoese the supremum problem of which is smaller than the radius of the ball of the respective node.

The authors provide a small set of experiments on one artificial and one real-world dataset in which the method consistently offers a speedup of around 40 times.

The paper is well written but not easy to follow, especially for someone who is not well-versed into optimization namely because it is quite technical consisting of a series of lemmas and theorems (just the appendix with the respective proofs is more than 20 pages).

Given the amount of the material and the NIPS space limitations I am not sure it would have been possible, even though desirable, to give a more high level picture/intuitive picture of the basic components of the algorithm before going into the technical details.

There seems to be a problem in the proof of Theorem 7. The definition of lambda max is in contradiction to ii) in that theorem, i.e. \lambda_max is the largest value that keeps y/\lambda in the feasible set, however in ii) we have y/\lambda \in F is equivalent to \lambda \geq \lamnbda_max.

Page 5 column 218 what is the \Ksi^i_j set?

On R1,2^* since these are supremums how often do we expect to see sup'>'w_j^i while the node is in fact inactive? any insights from the theory or experimental results?

Summary: A node screening method for hierarchical tree based regularization which brings an order of magnitude computational improvements.

Submitted by Assigned_Reviewer_3

The authors proposed a new approach in feature reduction for tree structured group lasso to reduce the complexity of the model by removing non-important features. The paper is written clearly and technically sounds correct. There are enough theoretical analysis to prove the effectiveness of the approach and experimental section has well in-depth analysis.

Overall, I think the contribution of the paper is significant. It has enough technical and experimental material to prove the idea.
Summary: Overall, I think the contribution of the paper is significant. It has enough technical and experimental material to prove the idea.

Submitted by Assigned_Reviewer_4

This paper proposed novel and generic principles for the Tree structured group Lasso (TGL). The proposed method uses novel hierarchical projections which flow a given tree structure. A key principle is the use of a condition which admits a closed form solution of the Lasso at the leaf of the tree. This enables an efficient computation of the solution. Another key is the use of the closed form solution of the nonconvex optimization problem on the supreme values of the dual optimum. These newly introduced principles enable high efficient and exact solving of the TGL problem.

The technical quality of the paper seems very high. The theoretical framework is well systematically provided. The explanation of the framework is mathematically well written, though the understanding these theories is quite hard within the limited explanation in this paper. But, this should be accepted because of the limited space of the paper. The two key principles proposed in this paper seem very original and play important role to achieve the significant performance of the TGL. The experiments demonstrate highly efficient performance of the proposed method in both artificial and real world data sets.

A drawback of this paper is the comparisons of the performance with the existing TGL methods. This can be added to the experimental results.

A minor comment What is H_G^i_j in Theorem 2? Define it.
Summary: This paper proposed a novel method of Tree structured group Lasso (TGL). The proposed principles seem to be very original and generic and to have high impact to the research field of structured regularization technique.

Author Feedback
Author rebuttal: We thank all reviewers for the constructive comments.

Reviewer 1

Q: The paper is well written but not easy to follow. It is desirable to give high level picture or intuitive picture of the basic components of the algorithm before going into the technical details.

A: Thanks for the comment. We will add more intuitions and discussions of the algorithms to the main text to improve the readability of the final version, if accepted.

Q: \lambda_max is the largest value that keeps y/ \lambda in the feasible set. However, in (ii) of Theorem 7, the feasibility of y/ \lambda is equivalent to \lambda>= \lambda_max.

A: To be consistent with existing literatures [21, 24, 26], \lambda_max is the largest value such that the solution is nonzero, namely, it is the largest value that keeps y/lambda outside of the feasible set. Thus, the definition of \lambda_max in this paper is consistent with (ii) of Theorem 7.

Q: What is the set \Xi^i_j in (R1*)?

A: In (R1^*), \Xi^i_j is an arbitrary set that contains [X^T\Theta]_{G_j^i} (\Theta is a set that contains the dual optimum). By the estimation technique in Section 5.1, we give one possible choice of \Xi^i_j in (18), which works very well as shown in the experiments.

Q: For (R1^*) and (R2^*), how often the ">" relationship holds for inactive nodes? Any insights from the theory or experimental results?

A: Figures 1 and 2 present the possibilities that the "<" relationship holds for inactive nodes, and we can see that they are very close to 1. Thus, the possibilities that the ">" relationship holds are very close to 0.

Reviewer 2

Q: The contribution of the paper is significant. It has enough technical and experimental material to prove the idea.

A: Thanks for the positive comment.

Reviewer 4

Q: The proposed principles seem to be very original and generic and to have high impact to the research field of structured regularization technique.

A: Thanks for the positive comment.

Q: A drawback of this paper is the comparisons of the performance with the existing TGL methods. This can be added to the experimental results.

A: Thanks for pointing this out. To the best of our knowledge, the proposed screening method, MLFre, is the first one that applies to TGL. Moreover, MLFre can be combined with any existing solver for TGL. In this paper, the solver is from the SLEP package [15], which is based on the accelerated gradient descent method. We will compare the speedup gained by MLFre combined with different solvers in the final version, if accepted.

Q: What is H_{G^i_j} in Theorem 2?

A: The definition is given in Notation; see the last paragraph in Section 1.

Reviewer 6

Q: In real-applications, how to build such a tree structure in the paper?

A: We adopt a similar fashion as in [13] to learn the tree structure. Specifically, we run the agglomerative hierarchical clustering on the correlation matrix of SNPs.

Q: How to guarantee this tree structure is really helpful to improve the performance?

A: In this paper, we focus on improving the efficiency of the TGL model. As we mentioned in the introduction, TGL performs very well for applications with hierarchical sparse patterns among the features. For this line of research, please refer to [10,12,13,14,18,27,28] and references therein.

Reviewer 7

Q: This paper presents an efficient algorithm for TGL. I think the content is interesting and the numerical result is convincing.

A: Thanks for the positive comment.